# Sedentary Behavior and Phase Angle: An Objective Assessment in Physically Active and Inactive Older Adults

**DOI:** 10.3390/nu16010101

**Published:** 2023-12-27

**Authors:** Liu-Yin Lin, Jiaren Chen, Ting-Fu Lai, Yen-Yu Chung, Jong-Hwan Park, Yih-Jin Hu, Yung Liao

**Affiliations:** 1Department of Health Promotion and Health Education, National Taiwan Normal University, Taipei 10610, Taiwan; 80905002e@ntnu.edu.tw; 2Zhongshan District Health Center, Taipei 10402, Taiwan; 3Department of Health, Long-Term Care Division, Taipei City Government, Taipei 11008, Taiwan; 4Graduate Institute of Sport, Leisure and Hospitality Management, National Taiwan Normal University, Taipei 106, Taiwan; 80931005a@ntnu.edu.tw (J.C.); 80905010e@ntnu.edu.tw (T.-F.L.); 61131027a@gapps.ntnu.edu.tw (Y.-Y.C.); 5Health Convergence Medicine Laboratory, Biomedical Research Institute, Pusan National University Hospital, Busan 49241, Republic of Korea; parkj@pusan.ac.kr

**Keywords:** nutrition, BIA, sedentary duration, cell health, accelerometer

## Abstract

The purpose of the present study was to investigate the associations of the objectively assessed total sedentary behavior (SB) amount and SB patterns with phase angle (PhA) in older populations and to conduct a comparison analysis for those who are physically active (meet 150 min/week) and inactive (less than 150 min/week). During May to August 2023, a convenience sampling was used to recruit older adults (aged ≥ 65 years) living in a community in Taiwan. The total SB amount (minutes/day), SB patterns (including SB bouts and breaks), and physical activity were assessed by a triaxial accelerometer. A multifrequency bioelectrical impedance analyzer (BIA) was used to measure PhA. Multiple linear regression analysis was employed to examine the associations between SB and PhA in the total number of participants and stratified in the physically active and inactive groups. For the whole sample (n = 166; mean age: 72.1 ± 5.5 years), the total SB amount and patterns of SB were not associated with PhA. For those physically active, the total SB amount and SB patterns were not associated with PhA. Among those physically inactive, the total SB amount was negatively related to PhA (B: −0.059, 95% CI: −0.118, 0.000). This study underscores the importance of minimizing the total SB amount in physically inactive older adults, providing evidence for future interventions targeting SB and PhA in this population.

## 1. Introduction

Globally, the population of older adults has increased significantly [1]. With age, physiological changes occur, including reduced metabolic rates and alterations in body composition, characterized by increased fat accumulation and decreased muscle mass. These changes lead to a heightened risk of chronic diseases and disabilities within this older demographic, imposing a substantial burden on global health and social care resources [2]. Nutrition is essential for the comprehensive physical and mental health of older adults. Adequate nutrition is essential for maintaining optimal bodily functions, preserving muscle mass, and ensuring healthy bone density, all of which become increasingly critical as individuals age.

Phase angle (PhA) is an emerging marker that reflects an individual’s cellular health and nutritional status [3,4]. There is a growing body of research that consistently demonstrates low PhA values associated with malnutrition [5] and increased nutritional risk [6], especially among populations affected by various diseases [7]. Furthermore, increasingly, recent research has shown that PhA is effective in screening for many aging-associated diseases, such as sarcopenia [8], frailty [9], and poor physical function [10]. Thus, PhA has been considered as a predictor of and an important outcome in assessing or reflecting the nutritional status and physical condition of the older population [11]. It is essential to identify the behavioral risk factors related to PhA in the daily routines of this demographic.

Sedentary behavior (SB) is a significant component of life behaviors and a crucial behavioral risk factor for many noncommunicable health conditions [12]. SB is defined as any waking activity with an energy expenditure of 1.5 metabolic equivalents (METs) or less, performed while sitting, reclining, or lying [13]. It is a well-established fact that sedentary time increases with age [14]. Studies have shown that objectively measured SB is notably prevalent among older adults against other age groups [15]. Previous studies have demonstrated that prolonged sedentary time is related to a heightened probability of diminished physical function [16], elevated mortality rates [17], and a greater likelihood of developing metabolic syndrome [15] in older adults. In addition to the overall SB amount, SB patterns (i.e., SB bouts and breaks) are also critical components of SB that are related to negative health consequences among older adults [18,19]. To our knowledge, only one prior study has explored the association between total SB time and PhA in the older population, without considering the potential association between SB patterns and PhA [20].

Concurrently, evidence from earlier research suggests that engaging in moderate-intensity physical activity at high levels may mitigate the elevated mortality risk related to prolonged SB [21,22]. The World Health Organization (WHO) has underscored the health advantages of physical activity (PA), recommending that older adults engage in a minimum of 150 min of moderate-to-vigorous physical activity (MVPA) each week [23]. Given that the health implications differ by meeting versus not meeting the WHO’s physical activity recommendations, understanding the differential impacts of SB on PhA based on whether people meet these recommendations is vital. However, the relationship of total SB amount and SB patterns with PhA in older adults according to whether they comply compliance with the WHO’s recommendations remain unclear. Therefore, this study focused on investigating the relationships of the objectively measured total SB amount and SB patterns (SB bouts and breaks) with PhA in Taiwanese older adults and compared these associations between physically active (≥150 min minutes/week MVPA) and inactive older adults (<150 min/week MVPA).

## 2. Materials and Methods

### 2.1. Participants and Study Design

This cross-sectional study was conducted between May and August 2023 in Taipei, Taiwan, and participants were recruited through neighborhood representatives (Figure 1). Eligible participants from the community were individuals aged 65 and above with the capacity for independent ambulation. Participants with artificial implants, such as cardiac pacemakers or joint replacements, that could interfere with the conduction of electrical currents and, thus, affect the bioelectrical impedance measurements were excluded from the study [24,25]. After providing their written informed consent, participants filled out a questionnaire detailing their sociodemographic information. They were then provided with a triaxial accelerometer that was worn on their waist for one week, except when engaging in water-based activities. Participants were asked to document any removal of the accelerometer along with daily sleep time. Bioelectrical impedance analysis (BIA) was utilized to assess body composition, including PhA. Upon completion of all of these steps, each participant was rewarded with a USD 7 gift voucher. Ethical clearance for the research was obtained from the Research Ethics Committee of National Taiwan Normal University (REC number: 202112HM024).

### 2.2. Measurements

#### 2.2.1. Objectively Measured SB

In this study, SB was quantified using a triaxial accelerometer (ActiGraph GT3X+, Pensacola, FL, USA), set to a sampling frequency of 30 Hz and 60 s epochs, defining sedentary time as periods with fewer than 100 counts per minute, which corresponds to an energy expenditures of ≤1.5 METs among older adults [26]. The total SB amount was calculated by summing the periods during which participants engaged in SB. A period of continuous sedentary time was categorized as an “SB bout”, whereas an “SB break” was categorized as a period of activity that interrupts sedentary time, occurring between consecutive SB bouts [13]. On the basis of a previous study, we quantified 10, 30, and 60 min of engagement in SB as SB bouts [27], and a nonsedentary activity between two SB bouts that lasted for 10 min as an SB break [28]. All these parameters were calculated using the accelerometer data.

The valid wear criteria were defined as follows [29,30]: a minimum of ≥600 min (10 h) of daily accelerometer wear time; at least four valid days of accelerometer wear, comprising three weekdays and one weekend day; nonwear time was marked by periods with a continuous zero count exceeding 60 min. The standard protocols presented in a past systematic review [30] for data collection and the processing criteria for data from accelerometers using ActiLife software (version 6.0, Pensacola, FL, USA) were followed in this study.

#### 2.2.2. MVPA According to the WHO’s Recommendation

MVPA was defined as any activity with energy expenditures ≥3 METs [31]. The duration of MVPA in this research was assessed via an ActiGraph GT3X+ device, operating at a 30 Hz sampling rate. The accelerometer cut-point for MVPA was set to ≥2020 counts/min [26]. According to the World Health Organization’s [23] guidelines on physical activity and sedentary behavior for older adults, we classified participants into two groups: those who were or were not achieving the 150 min MVPA per week. Participants who achieved 150 min MVPA weekly were in the physically active group, while those who did not were categorized as the physically inactive group.

#### 2.2.3. PhA

PhA was assessed by analyzing the primary resistance (R) and reactance (Xc) data through bioelectrical impedance analysis, employing a multifrequency bioelectrical impedance analyzer (MC-780MA, TANITA, Tokyo, Japan) [24]. This analyzer uses currents of various frequencies (5, 50, and 250 kHz) for the precise measurement of the body’s extracellular and intracellular fluids. Furthermore, measurements of PhA at a 50 kHz frequency have demonstrated utility for physiological–clinical studies and serve as reliable indicators of cellular integrity and overall physical health status [32,33]. PhA (°) was derived employing the following formula: reactance (Xc)/resistance (R) × 180°/π [32]. During the measurements, the participants stood on the analyzer’s metallic electrode footplates without shoes, assumed a neutral posture, and held the metallic handgrips with extended arms. Proper contact was ensured with all five fingers on the hand electrode and both the heel and forefoot on the circular foot electrode [34]. Moreover, participants were not allowed to wear any metal objects. Standard prediction equations, as opposed to athlete-specific equations, were applied to all participant measurements. All procedures were performed by well-trained staff. The results obtained using the device are reliable, reproducible, and show a strong correlation with conventional impedance equipment [35,36]. Data were exported and calculated using GMON software version 3.4.2 (Medizin & Service GmbH, Chemnitz, Germany).

#### 2.2.4. Covariates

This research accounted for various covariates including sociodemographic characteristics such as age, gender (male/female), current marital status (married/unmarried), residential status (alone or with others), level of education (below graduate level or graduate level or above), and current employment status. Body mass index (BMI) was calculated using BIA, with the following formula: weight, in kilograms, divided by the square of height, in meters. The average MVPA per day and accelerometer wear time were also included.

### 2.3. Statistical Analyses

Descriptive statistics analysis was utilized to illustrate the sociodemographic and other pertinent characteristics of the participants. The sociodemographic characteristics included age, gender, current marital status, residential status, current employment status, and level of education. Other pertinent characteristics included the proportion of physically active and inactive participants, BMI, average PhA value, SB patterns (number of ≥10, ≥30, and ≥60 min sedentary bouts; number of ≥10 min SB breaks), average MVPA per day, and accelerometer wear time. After adjusting for potential confounders, three multiple linear regression models were applied to explore the associations between the total SB amount and SB patterns (number of ≥10, ≥30, and ≥60 min sedentary bouts; number of ≥10 min sedentary breaks) and PhA among all participants. Furthermore, this analysis was stratified specifically in groups of physically active and inactive older adults. Total sedentary behavior time was divided by 30 min units. We computed unstandardized coefficients (B) along with their 95% confidence intervals (CIs) for the analysis. The statistical procedures were performed using SPSS Statistics software, version 27.0, and a *p*-value of less than 0.05 was adopted as the criterion for statistical significance.

## 3. Results

### 3.1. Sociodemographic Characteristics

We initially recruited 198 participants for the first phase of the study. However, after excluding incomplete questionnaires (n = 10), accounting for missing data (n = 15), and disqualifying individuals with artificial implants (n = 7), such as cardiac pacemakers or joint replacements, a total of 166 older adult participants (72.1 ± 5.5 years) were, ultimately, enrolled in the study. Table 1 summarizes the characteristics of the participants. In the total study sample, 72.3% were aged 65–74 years, 78.3% were women, 85.5% were married, 82.5% lived with others, 93.4% were unemployed, 59.6% had at least a university degree, and approximately 62% were physically inactive. The average BMI was 23.1, and the average of the whole body PhA measurements was at 5.0°. On average, participants engaged in SB for 578.5 ± 71.7 min/day, 15.4 ± 4.1 sedentary bouts lasting ≥10 min, 3.1 ± 1.7 sedentary bouts lasting ≥ 30 min, 0.6 ± 0.6 sedentary bouts lasting ≥60 min per day, and 15.2 ± 4.1 sedentary breaks lasting ≥10 min per day. The average MVPA per day and average accelerometer wear time was 21.4 ± 18.8 and 896.9 ± 67.1 min/day.

### 3.2. Relationships of Total SB Amount and SB Patterns with PhA in the Total Number of Participants

Table 2 indicates that, across all three models, no significant relationship was observed between the total SB time and the patterns of SB (including the number of ≥10, ≥30, and ≥60 min sedentary bouts and the number of ≥10 min sedentary breaks) with PhA for the total number of participants. In Model 1, accounting only for accelerometer wear time, no statistically significant association was found between total SB duration (B: −0.038, 95% CI: −0.079, 0.002) and PhA, nor between various SB patterns—such as the number of sedentary bouts lasting ≥10 min (B: 0.019, 95% CI: −0.009, 0.047), ≥30 min (B: 0.006, 95% CI: −0.054, 0.066), ≥60 min (B: −0.035, 95% CI: −0.202, 0.131), and sedentary breaks of ≥10 min (B: 0.019, 95% CI: −0.009, 0.047)—and PhA. Model 2, which further controlled for variables such as age, gender, BMI, and average MVPA, in addition to wear time, similarly showed no significant associations between total SB time (B: −0.033, 95% CI: −0.074, 0.008) and PhA, as well as between the detailed SB patterns and PhA, with the coefficients remaining statistically nonsignificant. Model 3, incorporated additional adjustments for the level of educational, marital status, residential status, and employment status. The results persisted with no significant relationship between total SB time (B: −0.038, 95% CI: −0.080, 0.003) and PhA, as well as between all SB patterns and PhA, with all *p*-values above the 0.05 threshold.

### 3.3. Relationships of Total SB Amount and SB Patterns with PhA in Physically Active and Inactive Participants

Table 3 depicts the results from three multiple linear regression models that stratified participants into physically active and inactive groups. In models 1 and 2, no significant relationship was observed between the total SB time and the patterns of SB (including number of ≥10, ≥30, and ≥60 min sedentary bouts and the number of ≥10 min sedentary breaks) with PhA among both physically active and inactive participants. However, in the subgroup of physically inactive participants, model 3—which accounts for a comprehensive range of variables including age, gender, BMI, educational level, marital status, residential status, employment status, average daily MVPA, and accelerometer wear time—revealed a significant association between the total amount of SB and PhA (B: −0.059, 95% CI: −0.118, 0.000). This indicates that an additional 30 min of daily SB is associated with a decrease of 0.059 units in PhA. Yet, the investigation of the relationships between various SB patterns and PhA shows no significant associations. This was consistent across the number of sedentary bouts lasting ≥10 min (B: 0.022, 95% CI: −0.013, 0.057), ≥30 min (B: 0.003, 95% CI: −0.075, 0.082), and ≥60 min (B: −0.135, 95% CI: −0.350, 0.079), as well as the number of sedentary breaks of ≥10 min (B: 0.022, 95% CI: −0.013, 0.057). Among the physically active participants in model 3, after adjusting for the same potential confounders, no significant relationships were found between total SB time (B: 0.010, 95% CI: −0.054, 0.074), the various SB bouts (the number of sedentary bouts lasting ≥10 min (B: 0.018, 95% CI: −0.034, 0.070), ≥30 min (B: 0.081, 95% CI: −0.015, 0.176), and ≥60 min (B: 0.228, 95% CI: −0.007, 0.464)), or sedentary breaks of ≥10 min (B: 0.018, 95% CI: −0.034, 0.069) and PhA.

## 4. Discussion

To our knowledge, this research appears to be a pioneer in assessing the relationship between total SB amount, SB patterns, and PhA using objective measures in community-dwelling Taiwanese older adults. The principal discovery of our study highlights the significant negative association between total SB amount and PhA among physically inactive older adults (i.e., not achieving 150 min/week of MVPA). However, in the analyses for the whole sample and exclusively for physically active older adults, PhA showed an association with neither total SB amount nor with SB patterns. Given that PhA serves as an effective instrument for nutritional screening in several diseases [4,11,25], the evidence from this study is informative for public health recommendations to incorporate strategies to reduce sedentary behavior duration among the physically inactive older adult population.

As aforementioned, when considering the whole sample, no associations were found for total SB amount and SB patterns with PhA. This observation aligns with the findings reported in prior research [20,37]. A possible reason for this finding may be that the findings for the physically active participants in the sample offset the impact of SB that occurs in physically inactive older participants [21]. One potential mechanism is that PA exerts a protective mechanism against the inflammatory response induced by prolonged sedentary time [38].

Among the physically inactive participants, there was a significant negative association between total SB amount and PhA. There is a physiological mechanism that can potentially explain this significant negative association, as follows: PhA serves as an indicator of cellular health and has been validated as a predictor of oxidative stress and inflammation in older adults [39]. Compared to physically active older individuals, those who are less active and spend extended periods sitting tend to exhibit elevated levels of oxidative stress biomarkers such as reactive oxygen species [40]. This prolonged SB duration can also trigger chronic systemic inflammation [41,42], which, in turn, can affect tissue hydration and permeability, potentially leading to hypoalbuminemia associated with malnutrition [7]. Both oxidative stress and inflammation can damage cellular structures and impair cellular functions, resulting in variations in PhA values [43]. Although previous studies have identified a relationship between SB patterns and outcomes, such as mortality [44] and physical function [45], in older populations, this study found no significant statistical relationship between SB patterns and PhA. Thus, despite the potential for speculation based on these pieces of evidence, the exact underlying physiological mechanisms remain unclear. The current literature, nonetheless, suggests that minimizing total SB time is crucial for enhancing PhA, particularly in physically inactive older adults. In summary, there is a need for more prospective studies to clearly establish the association between SB patterns and PhA in older population. Among those participants who were physically active, consistent with observations across the whole sample, no association was found between the extent of total sedentary behavior or its patterns and PhA. The potential underlying mechanism is likely similar to that for the whole sample, where increased PA could counteract the detrimental impacts of extended periods of being seated [38]. 

Nutrition emerges as a pivotal element of health in the older population, with PhA serving as a valuable tool for assessing their nutritional status [46]. Our study has found that, in addition to dietary behavior, SB is also related to PhA in physically inactive older adults. Therefore, it is imperative for public health interventions to consider sedentary duration when addressing nutritional outcomes in this physically inactive demographic. One of the strengths of this research lies in the employment of objective measures to assess both SB and PhA, which ensured accurate and robust data by minimizing potential errors or biases and ensuring the reliability and validity of the results. However, this study also had several limitations. First, it did not consider dietary intake on the day of measurement, which could influence PhA [7]. Subsequent research on PhA will include nutritional assessments and dietary questionnaires to account for dietary status. Second, given that the outcomes from BIA might differ based on the device utilized, some care is required when interpreting the findings. Efforts in future research should focus on harmonizing technology and ensuring the consistent calibration of bioelectrical impedance measurements. Third, this study used a cross-sectional design, which is not equipped to establish cause-and-effect relationships. Further studies employing longitudinal or interventional designs are necessary to confirm the causal relationships. Finally, a future study will recruit a more extended and representative sample to explore the association between SB and PhA among older adults while also analyzing differences by sex and age group.

## 5. Conclusions

These findings highlight the importance of reducing the total SB amount to enhance PhA in physically inactive older adults. These results provide preliminary evidence for future intervention studies on SB and PhA targeted at physically inactive older adults. Further studies using prospective design are needed to establish the directions of the causal relationships between SB and PhA.

## Figures and Tables

**Figure 1 nutrients-16-00101-f001:**
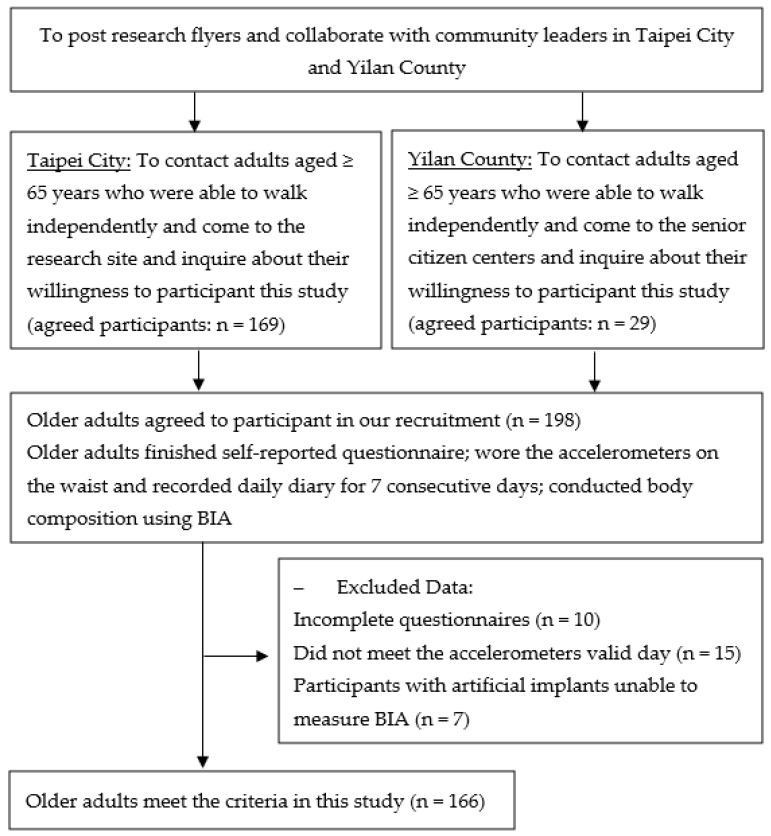
Flow chart of the participant recruitment procedure.

**Table 1 nutrients-16-00101-t001:** Characteristics of the total participants (n = 166).

Categorical Variables	*n*	%
Age		
65–74	120	72.3
≥75	46	27.7
Gender		
Female	130	78.3
Male	36	21.7
Current Marital Status		
No	24	14.5
Yes	142	85.5
Residential Status		
Alone	29	17.5
With others	137	82.5
Current Employment Status		
No	155	93.4
Yes	11	6.6
Education level		
No graduate level	67	40.4
Graduate and higher level	99	59.6
Met 150 min/week MVPA		
Physically active	63	38.0
Physically inactive	103	62.0
**Continuous variables**	**Mean**	**SD**
BMI (kg/m^2^)	23.1	3.3
The whole-body PhA (°)	5.0°	0.58
Sedentary behavior (min/day)	587.5	71.7
Number of ≥10 min sedentary bouts (times/day)	15.4	4.1
Number of ≥30 min sedentary bouts (times/day)	3.1	1.7
Number of ≥60 min sedentary bouts (times/day)	0.6	0.6
Number of ≥10 min sedentary breaks (times/day)	15.2	4.1
Average MVPA per day (min/day)	21.4	18.8
Accelerometer wear time (min/day)	896.9	67.1

BMI: body mass index; SD: standard deviation; MVPA: moderate-to-vigorous physical activity; PhA: phase angle.

**Table 2 nutrients-16-00101-t002:** The relationships between objectively measured total sedentary behavior amount and sedentary behavior patterns with phase angle in total participants (n = 166).

Variables	Model 1			Model 2			Model 3		
	B	95%CI	*p*	B	95% CI	*p*	B	95% CI	*p*
**Total sedentary behavior time**	−0.038	(−0.079, 0.002)	0.062	−0.033	(−0.074, 0.008)	0.112	−0.038	(−0.080, 0.003)	0.072
**Number of sedentary bouts (≥10 min)**	0.019	(−0.009, 0.047)	0.188	0.010	(−0.016, 0.036)	0.447	0.014	(−0.013, 0.041)	0.296
**Number of sedentary bouts (≥30 min)**	0.006	(−0.054, 0.066)	0.846	0.016	(−0.039, 0.072)	0.560	0.020	(−0.037, 0.077)	0.492
**Number of sedentary bouts (≥60 min)**	−0.035	(−0.202, 0.131)	0.675	0.016	(−0.138, 0.170)	0.839	0.008	(−0.147, 0.162)	0.921
**Number of sedentary breaks (≥10 min)**	0.019	(−0.009, 0.047)	0.191	0.010	(−0.016, 0.036)	0.453	0.014	(−0.013, 0.041)	0.300

B: unstandardized coefficient; CI: confidence interval. Model 1 controlled for accelerometer wear time. Model 2 controlled for age, gender, body mass index, average moderate-to-vigorous physical activity per day, and accelerometer wear time. Model 3 controlled for age, gender, body mass index, educational level, current marital status, residential status, current employment status, average moderate-to-vigorous physical activity per day, and accelerometer wear time. Sedentary bouts (≥10, ≥30, and ≥60 min) and sedentary breaks (≥10 min) were also adjusted for total sedentary behavior duration. Total sedentary behavior time was divided by 30 min units.

**Table 3 nutrients-16-00101-t003:** The relationships between objectively measured total sedentary behavior amounts and sedentary behavior patterns with phase angle in physically active (n = 63) and inactive participants (n = 103).

Variables	Active (n = 63)			Inactive (n = 103)		
	B	95%CI	*p*	B	95% CI	*p*
Model 1 adjusted for accelerometer wear time.						
Total sedentary behavior time	−0.012	(−0.076, 0.052)	0.711	−0.037	(−0.096, 0.021)	0.205
Number of sedentary bouts (≥10 min)	0.027	(−0.027, 0.082)	0.316	0.018	(−0.016, 0.053)	0.296
Number of sedentary bouts (≥30 min)	0.063	(−0.034, 0.159)	0.197	−0.01**7**	(−0.095, 0.061)	0.669
Number of sedentary bouts (≥60 min)	0.077	(−0.170, 0.324)	0.535	−0.115	(−0.340, 0.110)	0.312
Number of sedentary breaks (≥10 min)	0.027	(−0.027, 0.081)	0.323	0.018	(−0.016, 0.053)	0.298
Model 2 adjusted for age, gender, body mass index, average moderate-to-vigorous physical activity per day, and accelerometer wear time.						
Total sedentary behavior time	0.002	(−0.057, 0.060)	0.953	−0.053	(−0.110, 0.005)	0.072
Number of sedentary bouts (≥10 min)	0.013	(−0.036, 0.062)	0.589	0.018	(−0.016, 0.052)	0.307
Number of sedentary bouts (≥30 min)	0.063	(−0.025, 0.151)	0.157	0.006	(−0.071, 0.083)	0.875
Number of sedentary bouts (≥60 min)	0.200	(−0.021, 0.422)	0.075	−0.100	(−0.312, 0.112)	0.352
Number of sedentary breaks (≥10 min)	0.013	(−0.036, 0.062)	0.600	0.018	(−0.016, 0.051)	0.307
Model 3 adjusted for age, gender, body mass index, level of education, current marital status, residential status, current employment status, average moderate-to-vigorous physical activity per day, and accelerometer wear time.						
Total sedentary behavior time	0.010	(−0.054, 0.074)	0.751	−0.059	(−0.118, 0.000)	0.049 *
Number of sedentary bouts (≥10 min)	0.018	(−0.034, 0.070)	0.491	0.022	(−0.013, 0.057)	0.212
Number of sedentary bouts (≥30 min)	0.081	(−0.015, 0.176)	0.097	0.003	(−0.075, 0.082)	0.932
Number of sedentary bouts (≥60 min)	0.228	(−0.007, 0.464)	0.057	−0.135	(−0.350, 0.079)	0.214
Number of sedentary breaks (≥10 min)	0.018	(−0.034, 0.069)	0.501	0.022	(−0.013, 0.057)	0.213

B: unstandardized coefficient; CI: confidence interval. Sedentary bouts (≥10, ≥30, and ≥60 min) and sedentary breaks (≥10 min) were also adjusted for total sedentary behavior duration. Total sedentary behavior time was divided by 30 min units. * *p* < 0.05.

## Data Availability

Data and materials supporting the findings of this study are available upon reasonable request. Interested researchers can reach the corresponding author at liaoyung@ntnu.edu.tw for further details.

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
