# Peer review of "Sedentary Behavior and Phase Angle: An Objective Assessment in Physically Active and Inactive Older Adults"

_nutrients, 2023, doi:10.3390/nu16010101_

Round 1

Reviewer 1 Report

Comments and Suggestions for Authors

Clarity and Structure: The paper is well-structured with a clear abstract, introduction, methodology, results, discussion, and conclusion sections. However, the introduction could benefit from a brief explanation of why the study focuses on older adults, considering their unique physiological characteristics.

Methodology: The methods are thoroughly explained, particularly the use of a triaxial accelerometer and bioelectrical impedance analyzer. However, the selection criteria for participants could be more detailed, especially regarding the exclusion of individuals with artificial implants.

Results: The results are presented clearly, but the tables and figures could be improved for better clarity. Consider highlighting key findings in the tables or providing more descriptive captions.

Discussion: The discussion effectively ties the results to the broader context of sedentary behavior research. It would be beneficial to further explore the implications of the findings for public health recommendations for older adults.

Conclusion: The conclusion succinctly summarizes the study’s findings. It could be enhanced by suggesting specific directions for future research based on the study's limitations.

References: The references are appropriate and current. Ensure that all cited studies are relevant and add to the understanding of the topic.

Language and Grammar: The manuscript is well-written with minor grammatical errors. A careful proofreading by a native English speaker is recommended to refine language usage.

Ethical Considerations: The ethical considerations and participant consent are well addressed. Confirm that all participant data is handled in accordance with privacy regulations.

Overall, the manuscript presents valuable insights into the relationship between sedentary behavior and phase angle in older adults. With minor revisions, particularly in the explanation of participant selection and enhancement of tables and figures, the paper will be a significant contribution to the field.

Comments on the Quality of English Language

ok

Reviewer 2 Report

Comments and Suggestions for Authors

The study is methodologically sound and offers valuable insights into the relationship between sedentary behavior and cellular health, as indicated by PhA, in older adults. The differentiation between physically active and inactive individuals adds depth to the research, recognizing the varying lifestyles within the older population.

The use of objective measures for SB and PhA ensures accuracy and minimizes biases. However, the exclusion of dietary intake data, which can influence PhA, and the study's cross-sectional nature, which prevents causal inferences, are notable limitations. These aspects should be considered in future research to gain a more comprehensive understanding of the factors influencing PhA in older adults.

Overall, the study makes a significant contribution to the field of geriatric health and wellness, particularly in understanding the impact of lifestyle behaviors on cellular health markers. Its findings could be instrumental in developing targeted interventions to enhance the well-being of older adults, particularly those who are physically inactive.

Strengths and Limitations:

  • One of the study's strengths is the objective measurement of SB and PhA, enhancing the reliability of the data.
  • However, the study did not consider dietary intake, which might influence PhA.
  • The cross-sectional design limits the ability to establish causal relationships.
  •  
  •  

Reviewer 3 Report

Comments and Suggestions for Authors

General comments:

1. The paper presents an empirical study on sedentary behavior and phase angle with an objective assessment in physically active and inactive older adults.

2. The paper is well written and understandable.

3. The research topic is of relevance both for research and practice.

4. Important strengths include the thorough assessments and the relevance for daily life.

Specific comments:

6. How does the study add to the literature that is already existing?

7. What is the novelty in detail?

8. Which concrete gaps are addressed?

9. How can existing conceptual models be advanced in depth with these results?

10. The match of the paper to the journal could be strengthened. As far I can see, there is no data on nutrition discussed in depth.

11. The participants were convenience-sampled. Is the sample representative of the older population?

12. How has the accelerometer been calibrated?

13. The measurement properties of the accelerometer can be displayed more comprehensively.

14. The assessment of PhA could be explained in more detail.

15. The reliability and validity of PhA need to be illustrated more comprehensively.

16. Did findings vary differentially by sex (women vs men) and/or age group (young-old vs old-old)?

17. More references could be added to strengthen the discussion.

18. The limitations can be discussed in more detail.

19. The cost-effectiveness ratio of the expensive objective methods could be discussed in more detail.

20. The translation of the findings to real life implementation could be highlighted further.
